# Current Situation for Pharmacists in Japanese Veterinary Medicine: Exploring the Pharmaceutical Needs and Challenges of Veterinary Staff to Facilitate Collaborative Veterinary Care

**DOI:** 10.3390/pharmacy12060179

**Published:** 2024-11-29

**Authors:** Taisuke Konno, Hiroyuki Suzuki, Naoto Suzuki, Kouji Okada, Yosuke Nishikawa, Daisuke Kikuchi, Hitoshi Nakamura, Yuriko Murai

**Affiliations:** 1Division of Clinical Pharmaceutics, Faculty of Pharmaceutical Sciences, Tohoku Medical and Pharmaceutical University, 4-4-1 Komatsushima, Aoba-ku, Sendai 981-8558, Miyagi, Japan; hsuzuki@tohoku-mpu.ac.jp (H.S.); nakamura@tohoku-mpu.ac.jp (H.N.); y-murai@tohoku-mpu.ac.jp (Y.M.); 2Department of Pharmacy, Tohoku Medical and Pharmaceutical University Hospital, 1-12-1 Fukumuro, Miyagino-ku, Sendai 983-8512, Miyagi, Japan; naoto.suzuki@hosp.tohoku-mpu.ac.jp (N.S.); kokada@tohoku-mpu.ac.jp (K.O.); y.nishikawa@tohoku-mpu.ac.jp (Y.N.); d.kikuchi@hosp.tohoku-mpu.ac.jp (D.K.); 3Division of Clinical Pharmaceutics and Pharmacy Practice, Faculty of Pharmaceutical Sciences, Tohoku Medical and Pharmaceutical University, 1-12-1 Fukumuro, Miyagino-ku, Sendai 983-8512, Miyagi, Japan; 4Clinical Pharmacy Practice Center, Faculty of Pharmaceutical Sciences, Tohoku Medical and Pharmaceutical University, 4-4-1 Komatsushima, Aoba-ku, Sendai 981-8558, Miyagi, Japan; 5Division of Community Medicine and Pharmacy, Faculty of Pharmaceutical Sciences, Tohoku Medical and Pharmaceutical University, 4-4-1 Komatsushima, Aoba-ku, Sendai 981-8558, Miyagi, Japan

**Keywords:** veterinary pharmacist, veterinary pharmacy, community pharmacy, companion animal, pharmaceutical care

## Abstract

While veterinary pharmacy is common overseas, it has yet to gain traction in Japan. To clarify the actual situation in Japan, two surveys were conducted, targeting 4017 facilities (2000 pharmacies/drug stores, 2000 veterinary medical facilities, and 17 university-affiliated veterinary hospitals). We received 324 responses from pharmacy staff and 217 from veterinary hospital staff. Pharmacists were involved in drug therapy for companion animals either via direct discussion with veterinarians or by referrals for prescriptions from veterinarians (11 respondents). Veterinary staff and pharmacists showed a disparity regarding pharmacists’ involvement in companion animal healthcare. Veterinary staff wanted pharmacists to provide pharmaceutical guidance, perform audits, supply drug information, and prepare drugs. Strong demand (72.2% of the veterinary staff) existed for consultations with pharmacists regarding medication issues. Veterinary medicine has evolved to meet the demand for the advanced care of companion animals. Veterinary staff have diverse and substantial requests for pharmacists. Integrating the expertise of both veterinary staff and pharmacists may facilitate a team-based approach to veterinary medicine and improve the quality of care for companion animals. Tailoring approaches to suit Japan’s unique circumstances and creating a conducive environment for effective communication between pharmacists and veterinary staff are pivotal for advancing veterinary pharmacy in Japan.

## 1. Introduction

In Japan, there are more pets than children due to the declining birthrate and aging population. A 2023 survey by the Pet Food Association estimated a total of approximately 6.8 million dogs and 9.07 million cats in Japan, totaling 15.87 million. This exceeds the number of children under the age of 15, which is 14.35 million [1,2]. Furthermore, in veterinary medicine, companion animals are increasingly regarded as cherished family members, leading to a surge in societal demand for advanced medical care [3]. Veterinary medicine, especially small-animal clinical practice, has witnessed transformative changes in response to such a growing awareness.

In the United States, the first report on veterinary pharmacists was published in 1956, which focused on the use of antibiotics [4]. Since then, some countries, including the United States and the United Kingdom, have established the concept of veterinary pharmacy, whereby pharmacists play pivotal roles in dispensing medication [5], ensuring drug supply [6], and educating pet owners in veterinary settings [7]. Moreover, veterinary pharmacists significantly contribute to optimizing medication therapies, ensuring patient safety, and enhancing overall healthcare delivery in animal clinics and hospitals [8]. In contrast, Japan has only recently started providing medical care for companion animals [9], which differs greatly from the situation in other countries, where specialized care for companion animals has been operational since the 1950s. Despite the provisions of Article 23 of the Pharmacist Act in Japan stipulating that pharmacists can dispense medications based on prescriptions from veterinarians, comprehensive reports on such practices are scarce, leaving the actual extent of pharmacist involvement in veterinary care largely unknown.

Utilizing the Veterinary Drug Side Effects Database of the Japanese Ministry of Agriculture, Forestry, and Fisheries, we previously found room for improvement in the collection and utilization of drug information, research, and pharmaceutical management strategies to optimize drug therapy in small-animal clinical practice in Japan [10]. Based on this background, in this study, we aimed to clarify the state of pharmacy involvement in small-animal medical care in Japan and conduct a questionnaire survey study targeting community veterinary and pharmacy staff working in small-animal medicine.

## 2. Materials and Methods

### 2.1. Survey

Two separate questionnaire surveys were conducted using Google Forms with participants from community pharmacies (Appendix A; 26 questions) and animal hospitals (Appendix A; 31 questions). The survey items comprised three sections: (1) the respondents’ basic information, (2) the actual situation and pharmaceutical challenges related to pharmacists’ roles in veterinary medicine, and (3) the possibility of pharmacists’ involvement in promoting team-based veterinary care. The questionnaire used in this study was created based on previous research [11,12]. The survey was conducted between 1 May and 31 August 2023. All methods were performed in accordance with the Declaration of Helsinki. This study was approved by the Ethics Review Committee of the Tohoku Medical and Pharmaceutical University (Ethics Review Number: 2023-0-001).

### 2.2. Study Participants and Facilities

The study facilities were randomly selected based on the number of registered animal medical facilities, pharmacies, and drugstores in each prefecture using iTownpage (NTT Town Page Co., Ltd., Tokyo, Japan, https://itp.ne.jp/, accessed on 3 April 2023), the Japan Animal Hospital Association Animal Hospital Search Site (https://animal-hospital.jaha.or.jp/search/, accessed on 3 April 2023), and various search engines (Google: https://www.google.co.jp/, accessed on 3 April 2023; Yahoo! Japan: https://www.yahoo.co.jp/, accessed on 3 April 2023). The number of inquiry letters sent to each prefecture was determined based on the number of pharmacy facilities reported by the Ministry of Health, Labor, and Welfare and the number of animal medical facilities reported by the Ministry of Agriculture, Forestry, and Fisheries. Finally, the questionnaire was shared with 4017 facilities, including 2000 pharmacies/drug stores, 2000 veterinary medical facilities, and 17 university-affiliated veterinary hospitals.

### 2.3. Data Collection

Inquiries were sent to the target survey facilities along with QR codes leading to the relevant web questionnaire, and responses were collected through Google Forms. The number of respondents from each facility was not set, and each willing participant responded to the survey only once. All respondents provided online informed consent to participate through the questionnaire web page.

### 2.4. Statistical Analysis

Responses from pharmacists and veterinary staff were compared using the chi-square test, with the *p*-value cut-off for significance set to 0.05. All statistical analyses were performed using R ver. 4.2.3 (R Foundation for Statistical Computing, Vienna, Austria).

## 3. Results

### 3.1. Number of Responses and Background of the Respondents

The survey received 328 responses from pharmacies/drugstores and 220 responses from veterinary hospitals. Consent to use responses for research was obtained from 324 pharmacy/drug store staff and 217 veterinary hospital personnel. The percentage of valid responses obtained among request letters sent was 16.2% for pharmacies/drug stores and 10.9% for veterinary hospitals. In the survey targeting veterinary hospitals, most responses were from facilities in the Kanto region. Regarding affiliation, facilities with corporate management accounted for 114 responses (52.5%), those with individual management for 97 (44.7%), and university animal hospitals for 5 (2.3%), with responses from facilities with seven or more employees accounting for 59.5% of the total (Table 1). Responses were received from 4 of the 17 university-affiliated animal hospitals. The staff working in veterinary hospitals included 181 veterinarians, 34 companion animal nurses, and 2 pharmacists. Among veterinary staff, only a few individuals held multiple qualifications (Table 2). In the survey targeting pharmacies/drugstores, most responses were from facilities in the Kanto region (Table 3). By affiliation, responses from dispensing pharmacies accounted for 90.1% of the total, with facilities employing four to six staff members being the most common (36.7%). The majority of pharmacists possessed a license certified by the Japan Pharmacists’ Education Center. In terms of proximity, 104 (47.9%) veterinary hospitals had a pharmacy nearby, compared with 57 (17.6%) pharmacies with a veterinary hospital nearby.

### 3.2. Pharmacy Involvement in Veterinary Medicine and the Actual Situation of Veterinary Hospital Operations

First, we present an overview of the experiences of pharmacists. Experience with dispensing or compounding requests from veterinarians was reported by 3.4% of respondents (Figure 1a). Furthermore, 2.5% of respondents reported having dispensed or compounded medications for companion animals (Figure 1b); 27.5% and 4.0% (Figure 1c,d) of the respondents reported that they had consulted with patients and veterinarians, respectively, regarding medicines or drug therapy for companion animals. Examples of such instances are presented in Table 4, where medications for companion animals were dispensed or compounded based on prescriptions issued by veterinarians or after discussions and guidance regarding treatment. Additionally, 25.0% of pharmacists responded affirmatively when asked if they could dispense drugs prescribed by veterinarians, with the cost fully borne by the patients, similar to human prescriptions (Figure 2). Veterinarians played a central role in pharmaceutical and therapy-related tasks at veterinary hospitals, whereas companion animal nurses provided general support for clinical tasks (Figure 3). In two facilities with pharmacists, tasks such as dispensing medication, providing medication instructions to pet owners, handling pharmaceutical information, and providing pharmaceutical management guidance services were performed by pharmacists. Regarding the issuance of external prescriptions, only a small number of respondents stated that their facilities issued prescriptions (2.8%) or did so occasionally (5.0%; Figure 4a). The prevalence of companion animal prescription records was found to be low; however, approximately 20% of respondents among veterinary staff reported that pet owners brought companion animal prescription records, known as “Okusuri-Techo”, to their facilities (Figure 4b). The issuance of pharmaceutical information documents was limited, with only 9.7% of respondents indicating that they issued them to all clients and 18.4% of respondents stating that they issued them only to those who requested them, accounting for approximately one-quarter of the total documents issued (Figure 4c). Pharmacists in veterinary hospitals were scarce, with only a few facilities reporting pharmacists (2.3%) or practitioners with dual licenses (3.7%; Figure 4d).

### 3.3. Perception of Pharmacists and Veterinary Hospital Staff Regarding Pharmacists’ Involvement in Companion Animal Healthcare

Among the veterinary hospital staff, 57.1% of respondents reported that they considered pharmacists’ involvement in companion animal healthcare, showing a statistically significant positive association, with adjusted residuals in the chi-squared test of 2.68, exceeding 1.96 at the 5% significance level (Table 5). Conversely, 45.4% of pharmacists working in pharmacies indicated that they had considered such involvement, demonstrating a statistically significant negative association, with adjusted residuals of −2.68. Pharmacists working in pharmacies and drugstores expressed a preference for collaboration to promote team veterinary care, with 114 respondents indicating a desire for collaboration, showing a statistically significant positive association with adjusted residuals of 2.07, exceeding 1.96 at the 5% significance level (Table 6). Conversely, 15 and 55 respondents among staff working in veterinary hospitals stated that they would not consider collaboration, with adjusted residuals of 2.51 and 3.37, respectively, showing a statistically significant positive association.

### 3.4. Pharmacists’ Perceptions of the Pharmaceutical Needs of Veterinary Staff and Their Contribution to Veterinary Medicine

The top three tasks that veterinary staff would like to delegate to pharmacists and their perceived importance were pharmaceutical/drug management guidance services, audit, and drug information for veterinary staff (Figure 5). The least important task involved mixing and preparing injections and infusions. Half of the veterinary hospital respondents (49.3%) reported experiences of uncertainty or difficulty in discernment during routine clinical practice, such as the evaluation of drug interactions (Figure 6a). Needs expressed by 25.8% of veterinary staff respondents included the widespread availability of pharmacies capable of dispensing prescriptions from animal hospitals and an understanding of pharmacist tasks specific to veterinary care, among others (Figure 6b).

The top three tasks in which pharmacists were most likely to participate in small-animal clinical practice were preparing the drugs with a prescription (excluding injectable drugs), audit, and drug management. Therapeutic drug monitoring was the least likely task. Some discrepancies were observed between the professions in terms of their perceptions of the work to which they could contribute and their needs (Figure 5).

### 3.5. Pharmacists’ and Veterinary Staff’s Attitudes Toward Problem-Solving and Barriers Faced by Pharmacists Engaging in Veterinary Care

The survey gathered information on whether veterinary staff consulted pharmacists when faced with pharmacological questions or challenges. Among the 217 veterinary staff members surveyed, 71.9% of participants responded affirmatively, 13.4% of participants responded negatively, and 14.3% of respondents were undecided, indicating that over two-thirds of veterinary staff reached out for pharmacy guidance (Figure 7a). Subsequently, the pharmacists’ responses when asked about pharmaceutical questions or challenges were investigated. In total, 11.7% of pharmacists reported assessing the question or consultation and responding only if they could immediately provide an answer; 65.5% of respondents responded within their knowledge scope, including questions they could not immediately answer, and consulted with others for answers while respecting their expertise; a few respondents preferred not to answer regardless of the query (Figure 7b).

Veterinary staff were surveyed regarding the presence of barriers when pharmacists participated in veterinary care. Overall, 6.9% of respondents reported no barriers, 30.4% of respondents were unsure, and 136 individuals (62.7%) acknowledged barriers (Figure 8a). The top barriers included high personnel costs associated with pharmacist employment, lack of pharmacist knowledge of animals, and insufficient productivity to justify the high personnel costs, collectively representing 80.5% of all responses (Figure 8a). Similarly, 9.0% of pharmacists reported no barriers, 56.2% of pharmacists were unsure, and 34.9% of respondents reported barriers. The main barriers included a lack of knowledge of animals (Figure 8b).

### 3.6. Pharmacists’ Approaches to Gaining Veterinary Knowledge

We investigated the perspectives of veterinary staff and pharmacists on how pharmacists should acquire knowledge of veterinary medicine and animals when engaging in small-animal clinical practice. The top three methods deemed appropriate by veterinary staff for acquiring knowledge were as follows (based on the number of affirmative responses): gaining clinical experience at an animal hospital while acquiring the necessary knowledge, with 162 responses (85.3%); progressing in learning and knowledge acquisition under the guidance of veterinarians, with 161 responses (85.2%); and participating in veterinary medicine-related conferences, seminars, and so on, with 156 responses (83.4%; Figure 9a).

In a similar question asking pharmacists how they believe they should acquire knowledge, the top three responses were as follows: not sure where to start studying, with 261 responses (83.4%); participating in veterinary medicine-related conferences, seminars, and so on, with 222 responses (73.3%); and progress in learning and knowledge acquisition under the guidance of veterinarians, with 215 responses (70.7%). These results indicated a disparity in the perception of learning methods between the two professions (Figure 9b).

## 4. Discussion

To the best of our knowledge, this is the first study to report the current state of veterinary pharmacy in Japan and the specific needs within Japanese small-animal clinical practice. The survey results revealed a clear recognition of the need for mutual cooperation between pharmacists and veterinary staff to optimize pharmacotherapy for companion animals in Japan, indicating an interest in collaboration. Nevertheless, a nuanced discrepancy was identified between the perceptions and needs of veterinary staff and pharmacists regarding the involvement of pharmacists in veterinary medicine. Reports from the United States (which is considered an advanced nation in veterinary pharmacy) indicate that dispensing medications to the owners of companion animals at community pharmacies brings maximum benefits to animals and their owners [13]. Bridging this gap in Japan could potentially lead to the wider adoption of team-based veterinary medicine, enhancing the overall quality of care for companion animals.

The survey results highlight the complexity of the current situation in Japan, wherein the role of pharmacists in veterinary medicine is not well defined. The academic field of veterinary pharmacy has a relatively short global history and has developed uniquely in accordance with the specific circumstances of each country. The treatment of companion animals is often considered to fall beyond the scope of pharmacists’ responsibilities in Japan. Additionally, veterinary medicine in Japan sees a low rate of private pet insurance coverage and is largely practiced under a system of medical treatment at the individual’s own expense without standardized pricing [14], further highlighting the conventional practice of veterinarians and veterinary nurses primarily preparing medications at animal hospitals. In addition to compliance concerns, the excessive load of various tasks on veterinarians and veterinary nurses can lead to a decline in job type specialization and operational inefficiency. Considering these findings, there is potential to introduce a division of labor and task shifting in veterinary medicine. We previously reported that there is scope for improvement in pharmaceutical management [10], which is consistent with the needs of veterinary staff for pharmacists identified in this study. Therefore, pharmacists may create added value and contribute to improving the quality of medical care by actively engaging in drug management guidance, pharmaceutical management, audit, and drug information, which are services that are commonly requested by veterinary medical staff. In particular, collaboration between community pharmacies and veterinary hospitals to support the evaluation of drug interactions and the utilization of drug information, which have been highly sought after by veterinary medical staff, has the potential to greatly expand treatment options and foster mutual understanding between experts in both fields. To achieve smooth collaboration, these pharmacist roles should be closely aligned with real-world veterinary clinical practice. Veterinary education for pharmacists increases their knowledge of the treatment of companion animals and promotes interprofessional collaboration [15]. However, in contrast to other countries, where educational programs for pharmacists in veterinary medicine [16,17] and professional associations [18,19] are available, Japan lacks such programs in its pharmaceutical science faculties and organizational structures [20]. Hence, establishing a foundation for a new academic field, developing educational curricula and training programs, and creating platforms are all necessary to facilitate collaboration and clarify the potential roles that pharmacists should play in small-animal clinical practice. A thorough examination of potential options in accordance with Japan’s circumstances and the tailoring of approaches are imperative to meet the unique needs of companion animals. Establishing an environment in which pharmacists can propose and implement suggestions for veterinary staff is crucial to navigating this complex situation.

The veterinary services provided in pharmacies currently remain limited to inquiry responses owing to the lack of advancement in the separation of drug prescription and dispensing activities. Promisingly, some pharmacists in this study reported involvement in companion animal pharmacotherapy in collaboration with veterinarians (Table 4). Examples like these embody merely a portion of the professional abilities and skills of pharmacists; they have the potential to yield positive effects in multiple aspects. The potential for community pharmacy development in Japan is demonstrated by the state of veterinary pharmacy in other countries. In Malaysia, which is in the same Asian region as Japan, pharmacies contribute to pet health management by supplying and dispensing medications to both veterinarians and pet owners. Additionally, in New Zealand, pharmacists collaborate with zoo veterinarians to improve the health of exotic animals [21,22]. These examples show that envisioning community pharmacies as appropriate healthcare facilities for veterinary medicine and establishing cooperation and collaboration between specialized professionals, including pharmacists and medical experts from multiple disciplines, can improve veterinary pharmacy in Japan. For example, by tailoring international practices related to drug dispensing and safety concerns [23] to suit specific Japanese conditions, opportunities can be created to optimize drug therapy by formulating medicines optimized for companion animals in forms that are easy for owners to administer. Such collaboration may also contribute to reducing the financial burden on pet owners and improving convenience. However, owing to the limited number of pharmacists involved in veterinary medicine and the dispersed nature of their employment in Japan, pharmaceutical knowledge remains fragmented, impeding the accumulation and transfer of knowledge and skills. Consequently, establishing an academic foundation and standardizing the educational system becomes challenging. Given that animal hospital staff may lack specific pharmaceutical information regarding different companion animals, pharmacists, and researchers play a crucial role in creating and disseminating useful drug-related information and basic data for therapy.

Despite the valuable insights it provides, this study does have some limitations. Responses from 383 pharmacists and 380 veterinarians were required to obtain optimal detection power. However, although the number of responses from pharmacists was close to ideal, neither group reached the target value, and statistical power was not achieved. Additionally, although the survey was conducted randomly, it is possible that the response rate was high owing to positively biased perceptions of pharmacist employment in compliance with dispensing rights, particularly in larger-scale animal hospitals.

The involvement of pharmacists in the healthcare of companion animals implies a proactive contribution to developing a unified “One Health” approach that aims to protect human, animal, and environmental health as an integrated whole [24]. Future research should aim to inform policies that facilitate the smooth integration of these two previously disparate fields. Such initiatives will enhance mutual understanding among professionals and contribute to the advancement of the academic field of veterinary pharmacy in Japan. Future research investigating the advantages, effectiveness, and changes in profitability resulting from the establishment of cooperative relationships between community pharmacies and animal hospitals would be beneficial. In Japan, the current priority for pharmacy optimization is focused on addressing major issues affecting human clinical medicine, such as the uneven regional distribution of pharmacists, uneven distribution among industries, decreasing working population, and limited finances; veterinary pharmacy, as an emerging field, should be considered in the future.

## 5. Conclusions

Although still rare in Japan, some pharmacists fill prescriptions for companion animals, handle inquiries from veterinarians, and work in veterinary clinics, indicating the involvement and potential role of pharmacists in small-animal medical care. The needs of veterinary clinic staff for pharmacists vary widely, with tasks such as providing information on drug interactions. However, a discrepancy was observed between these needs and the tasks that pharmacists believe they could handle. Approximately 50% of pharmacists working in pharmacies reported that they wish to be involved in veterinary medicine in the future but are uncertain about where to start learning. Promoting collaboration between veterinary staff and pharmacists for small-animal care offers support to the emerging field of veterinary pharmacy in Japan.

## Figures and Tables

**Figure 1 pharmacy-12-00179-f001:**
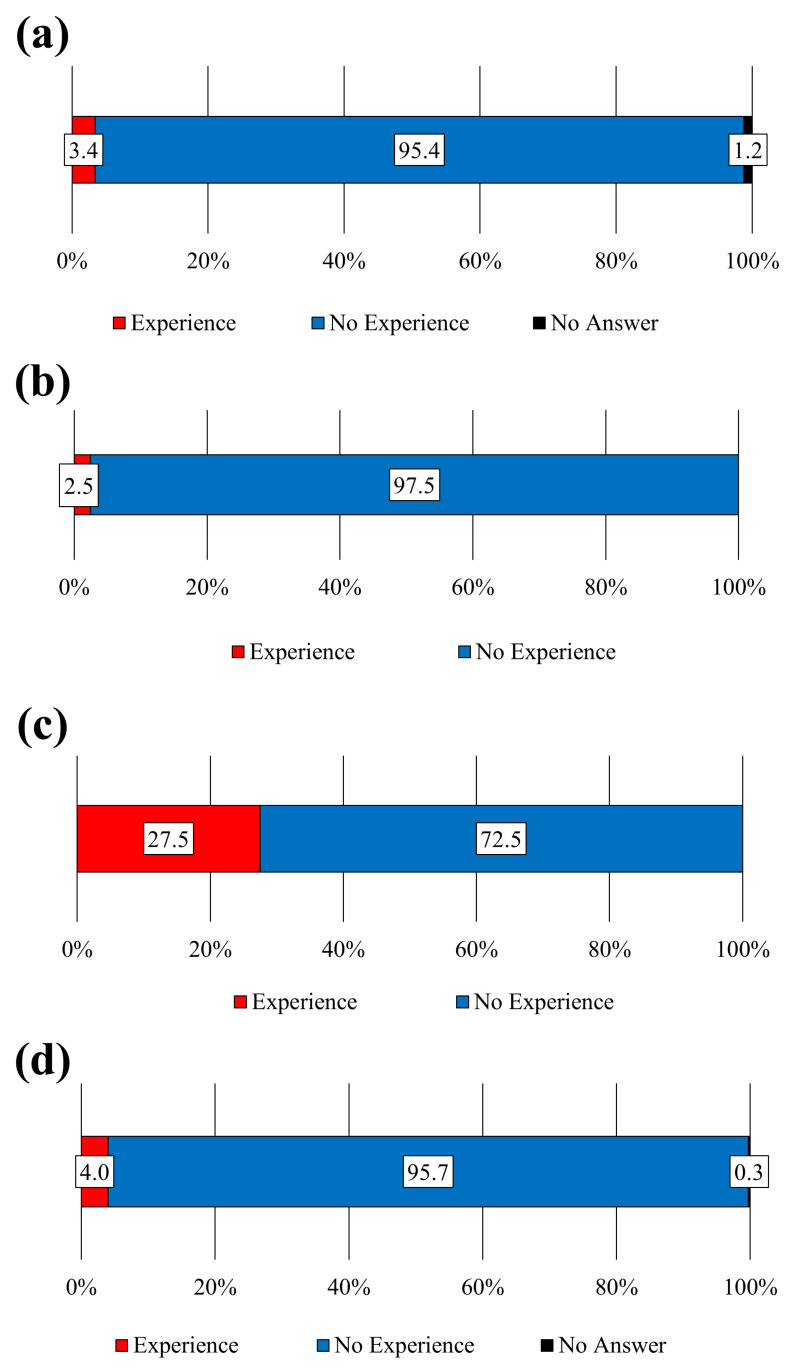
Presence or absence of dispensing experience at the request of a veterinarian and consultation on veterinary medicine. (**a**) Experience with requests from veterinarians for dispensing/preparing drugs (N = 324). (**b**) Experience dispensing and/or preparing medication at the request of a veterinarian (N = 324). (**c**) Experience consulting patients regarding companion animal medicines/drug therapy/health conditions (N = 324). (**d**) Consultation experience from veterinarians regarding companion animal medicines/drug therapy/health conditions (N = 324).

**Figure 2 pharmacy-12-00179-f002:**
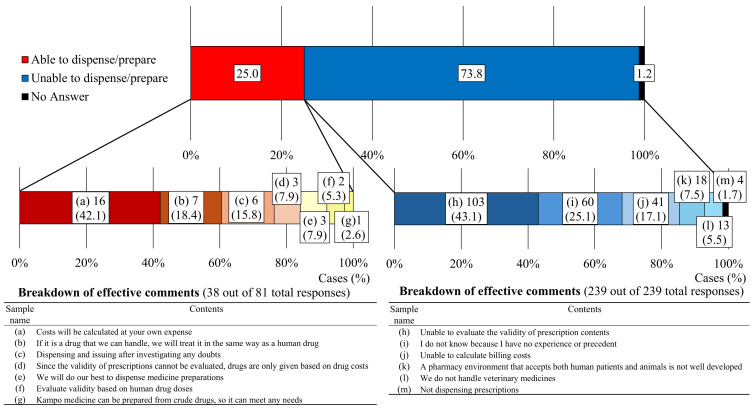
Responses to whether a drug can be dispensed after receiving a prescription from a veterinarian and reasons for these responses (N = 324).

**Figure 3 pharmacy-12-00179-f003:**
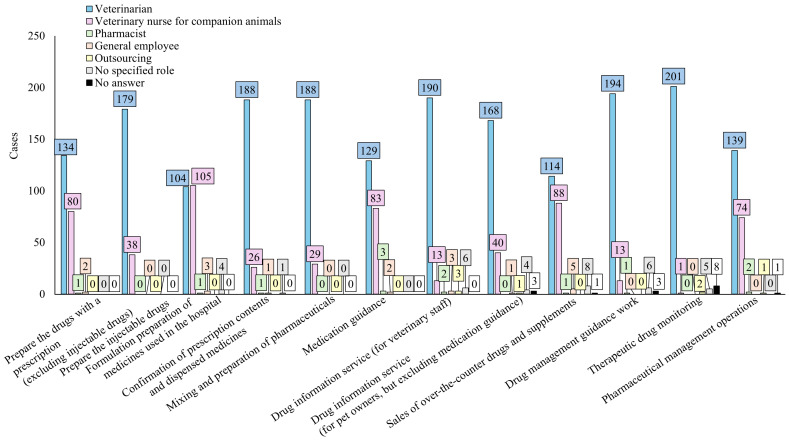
Positions in charge of pharmaceutical-related work at veterinary hospitals. N = 217 for each content.

**Figure 4 pharmacy-12-00179-f004:**
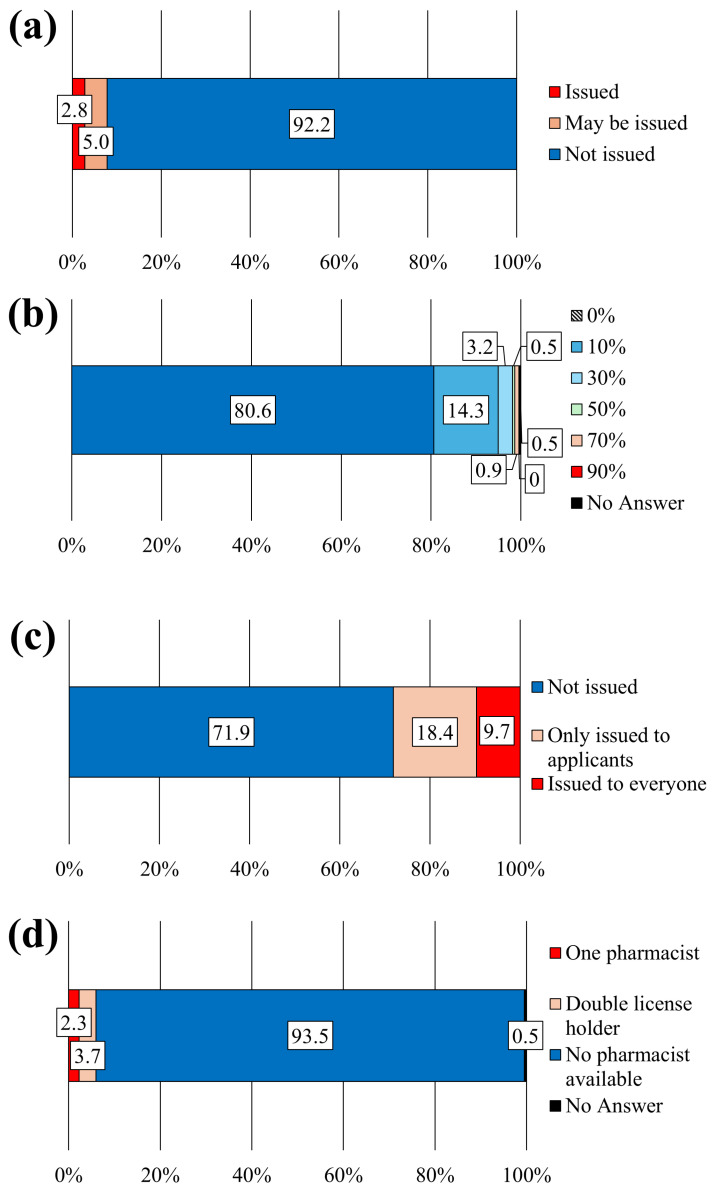
Use of out-of-hospital prescriptions, medicine notebooks, drug information documents, and enrollment status of pharmacists at veterinary hospitals. (**a**) Out-of-hospital prescription issuance status (N = 217). (**b**) Percentage of owners who bring their companion animal’s medicine notebooks (N = 217). (**c**) Distribution of drug information documents (N = 217). (**d**) Enrollment status of pharmacists in the animal hospital (N = 217).

**Figure 5 pharmacy-12-00179-f005:**
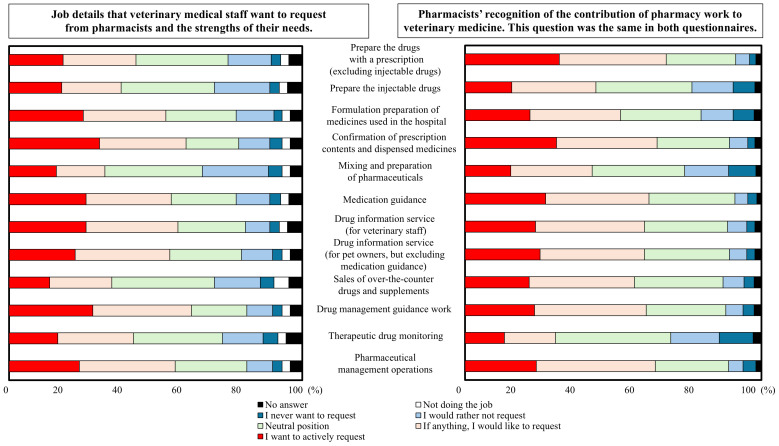
Needs of veterinary staff and degree of contribution of pharmacists. N = 217 for veterinary hospital staff. N = 324 for community pharmacy.

**Figure 6 pharmacy-12-00179-f006:**
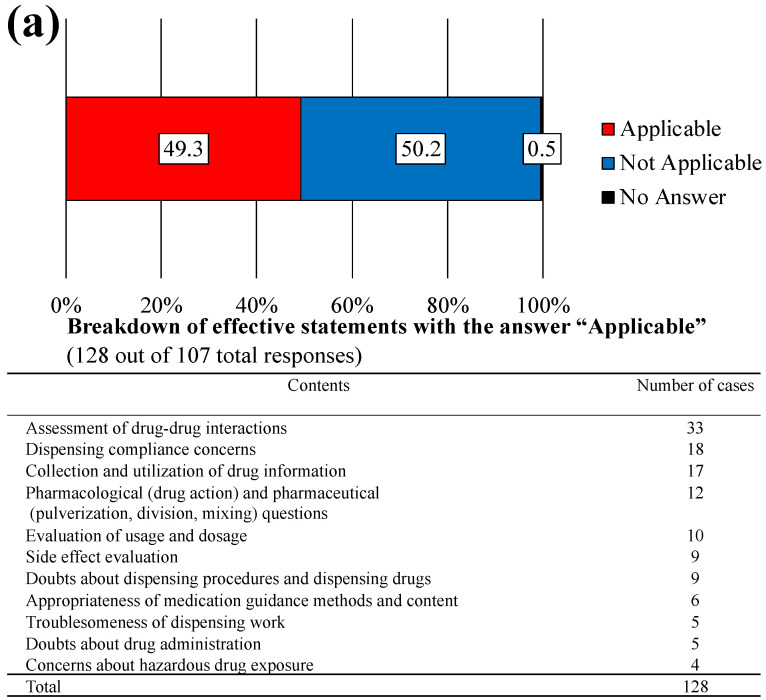
Pharmaceutical clinical questions and requests for pharmacists/pharmacies at veterinary hospitals. (**a**) Assessment of whether veterinary staff have any problems, doubts, or concerns, do not understand, or are at a loss of judgment with pharmaceuticals or drug therapy (N = 217). (**b**) Responses to the question: “As a professional working at a veterinary facility, do you have any requests or hopes for pharmacists or pharmacies?” (N = 217).

**Figure 7 pharmacy-12-00179-f007:**
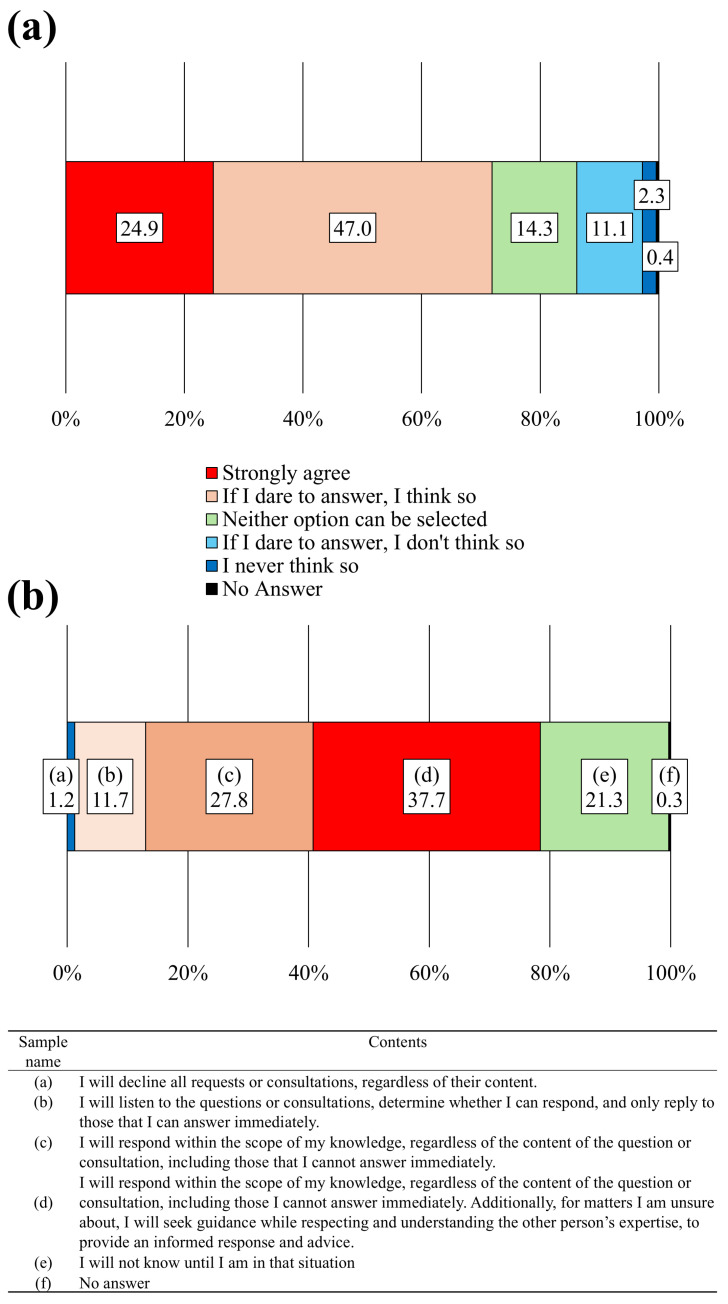
Attitudes toward problem-solving among veterinary medical staff and pharmacists. (**a**) Responses to the question: “When you have questions or concerns about medicines, would you like to feel free to ask or consult a pharmacist?” (N = 217). (**b**) Responses to the question: “What is your stance as a pharmacist when responding to questions or consultations from veterinary medical staff?” (N = 324).

**Figure 8 pharmacy-12-00179-f008:**
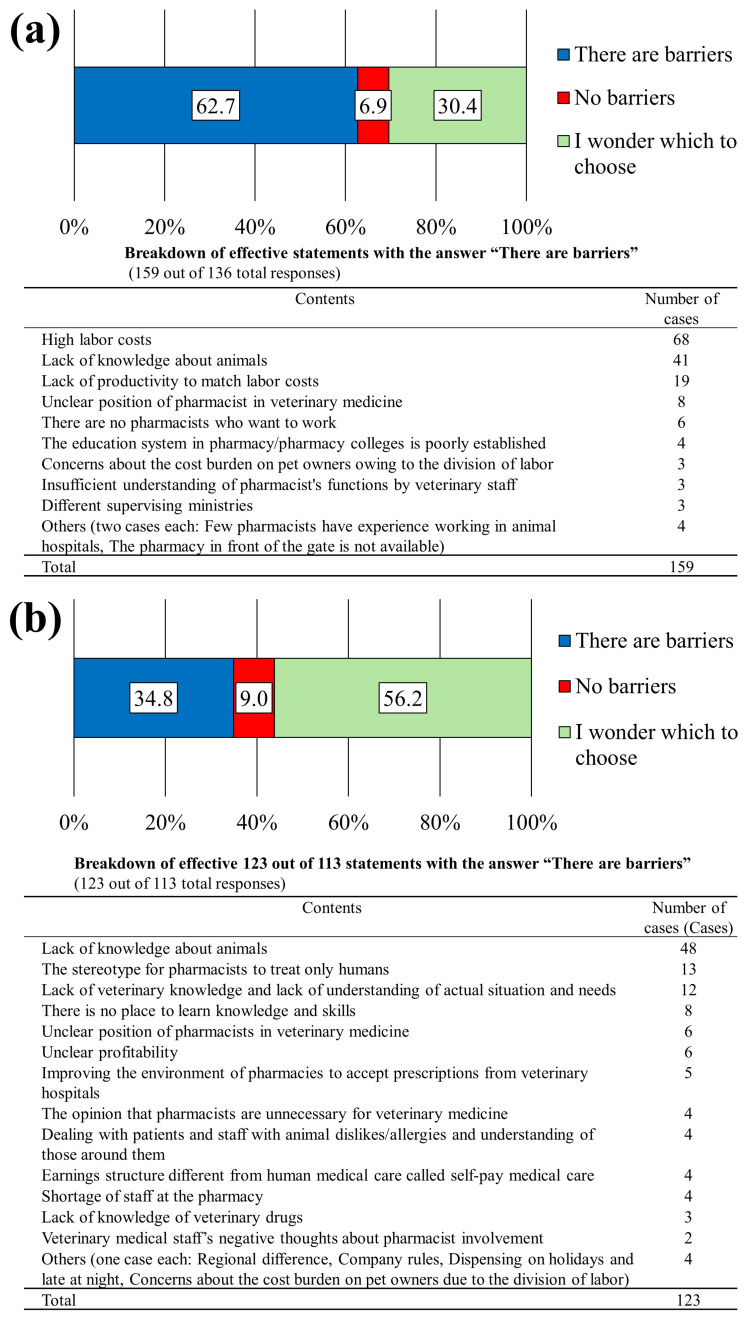
Existence and details of barriers to participation of pharmacists in veterinary medicine. (**a**) Responses from animal hospital staff (N = 217). (**b**) Responses from pharmacy/drug store pharmacists. This question was the same for both questionnaires (N = 324).

**Figure 9 pharmacy-12-00179-f009:**
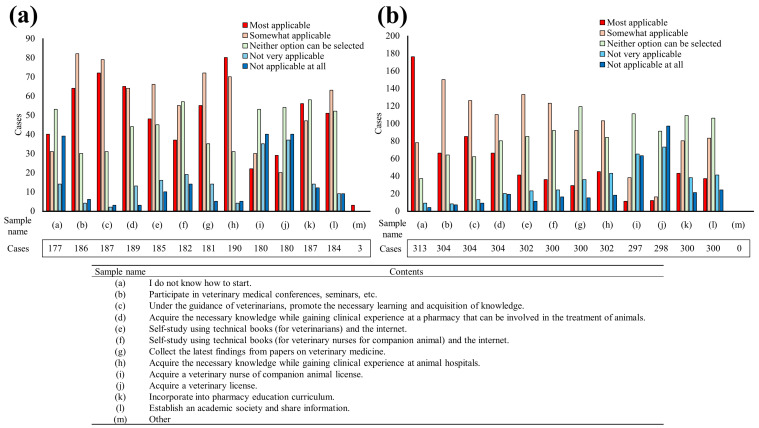
Perceptions of how the pharmacists involved in veterinary medicine should acquire specialized knowledge. (**a**) Responses from animal hospital staff. (**b**) Responses from pharmacy/drug store pharmacists. This question was the same for both questionnaires.

**Table 1 pharmacy-12-00179-t001:** Respondent characteristics for animal hospital staff.

Characteristic		N (%)	Total
Types of professionals	Veterinarian	181 (83.4)	217
	Veterinary nurse for companion animals	34 (15.7)	
	Pharmacist	2 (0.9)	
Region	Hokkaido	11 (5.1)	217
	Tohoku	21 (9.7)	
	Kanto	83 (38.3)	
	Chubu	30 (13.8)	
	Kinki	38 (17.5)	
	Chugoku/Shikoku	17 (7.8)	
	Kyushu/Okinawa	15 (6.9)	
	No answer	2 (0.9)	
Affiliation	Personal business	97 (44.7)	217
	Corporate management	114 (52.5)	
	University hospital	5 (2.3)	
	No answer	1 (0.5)	
Number of employees	1–3	32 (14.7)	217
	4–6	55 (25.3)	
	7–9	39 (18.0)	
	10–20	51 (23.5)	
	≥21	39 (18.0)	
	No answer	1 (0.5)	
Status of neighboring facilities	There is a pharmacy nearby	104 (47.9)	217
	There is NO pharmacy nearby	112 (51.6)	
	No answer	1 (0.5)	

**Table 2 pharmacy-12-00179-t002:** Qualifications of respondents employed in animal hospitals, pharmacies, and drug stores.

Characteristic			N (%)	Total	
Animal hospital staff	Qualifications other than veterinarian license	Pharmacist	7	22	
		Japanese Animal Hospital Association-certified veterinarian	6		
		Veterinarian certified by various societies	4		
		Doctor of veterinary medicine	1		
		Narcotics license for veterinarians	1		
		Acupuncturist	1		
		Public health laboratory technologist	1		
		Registered salesclerk	1		
	Qualifications other than veterinary nurse for companion animal license	Pharmacist	1	3	
		Medical technologist	1		
		Japan Kennel Club-certified pet nurse	1		
	Qualifications other than pharmacist’s license	-	0	0	
Pharmacy and drug store staff	Medical qualifications other than pharmacist’s license	Licensed by the Japan Pharmacists’ Education Center	43		75
		Sports pharmacist	7		
		Practical training certified guidance pharmacist	6		
		Medical technologist	4		
		Supplement advisor	3		
		Chinese herbal medicine certified pharmacist	2		
		Public health laboratory technologist	2		
		Care manager	2		
		Outpatient cancer treatment certified pharmacist	1		
		Regional pharmacy care pharmacist	1		
		Traditional Chinese physician	1		
		Certified diabetes educator in Japan	1		
		Certified infertility counselor	1		
		Registered salesclerk	1		
	Qualifications and certifications held in the field of veterinary medicine	Veterinarian	1		5
		Pet care worker	1		
		Type 1 registration of animal handling business	1		
		Pet breeding manager, second grade	1		
		Japanese Animal Hospital Association-certified family dog instructor	1		

**Table 3 pharmacy-12-00179-t003:** Respondent characteristics for pharmacy/drug store staff.

Characteristic		N (%)	Total
Region	Hokkaido	16 (4.9)	324
	Tohoku	37 (11.4)	
	Kanto	89 (27.5)	
	Chubu	42 (13.0)	
	Kinki	37 (11.4)	
	Chugoku/Shikoku	61 (18.8)	
	Kyushu/Okinawa	42 (13.0)	
Affiliation	Pharmacies (whose main business is dispensing prescriptions)	292 (90.1)	324
	Kampo pharmacies	17 (5.3)	
	Pharmacies (other than those listed above)	2 (0.6)	
	Drug stores	9 (2.8)	
	Companies handling veterinary drugs	1 (0.3)	
	No answer	3 (0.9)	
Number of employees	1–3	63 (19.5)	324
	4–6	119 (36.7)	
	7–9	60 (18.5)	
	10–20	65 (20.1)	
	≥21	5 (1.5)	
	No answer	12 (3.7)	
Status of neighboring facilities	There is an animal hospital nearby	57 (17.6)	324
	There is NO animal hospital nearby	265 (81.8)	
	No answer	2 (0.6)	

**Table 4 pharmacy-12-00179-t004:** Conditions for pharmacists dispensing medicines for a companion animal under the direction of a veterinarian.

Affiliation	Region	Specific Comments
Community pharmacy	Tohoku	At the owner’s request, ursodeoxycholic acid was dispensed as directed by a veterinarian. Medication instructions were given by the veterinarian.
Community pharmacy	Tohoku	An owner visited the pharmacy after receiving an out-of-hospital prescription. Although it was difficult to evaluate the prescription content, there was no significant deviation from the human dose. The cost of dispensing was all paid by me. It seemed that the cost was lower compared to that at the animal hospital.
Kampo pharmacies	Tohoku	After consulting with a veterinarian, the owner came to the pharmacy. Kampo-Hochuekkito and Kampo-Rikkunshito were prepared to reduce the side effects of chemotherapy.
Community pharmacy	Kanto	Dispensed prescription drugs for pet dogs.
Companies handling veterinary drugs	Kanto	I have experience working as a pharmacist in a veterinary hospital. If there was any doubt about the prescription, we confirmed and consulted with a veterinarian and prepared it in the hospital. I also gave the owner instructions on how to use the medication.
Kampo pharmacies	Kanto	There was a consultation/request from an animal hospital one station away. After mutual consideration, a Chinese herbal medicine was dispensed. Medication histories were recorded at both facilities. The expenses were billed to the animal hospital.
Kampo pharmacies	Chubu	A nearby veterinary hospital asked us to provide an eye drop (antibacterial drug) that was not used in the hospital. Prescription details were communicated to the veterinarian in advance. The owner brought the issued out-of-hospital prescription. The weight of the pet was replaced with that of a human to evaluate the adequacy of the prescription. Medication history management and medication counseling were given to the owner in the same manner as in the case of humans.
Community pharmacy	Chubu	Dispensed at the request of the owner who brought the out-of-hospital prescription. I could not provide proper medication counseling owing to my lack of knowledge about animals.
Pharmacies (other than those listed above)	Chugoku/Shikoku	The pharmacist consulted with a veterinarian about the medicine for a pet dog and dispensed it.
Community pharmacy	Kyusyu/Okinawa	The pharmacist had experience working at a veterinary facility. The pharmacist oversaw the delivery of veterinary medicines, inspection, filing, and medicine management.

**Table 5 pharmacy-12-00179-t005:** Consideration by pharmacists and veterinary medical staff of the involvement of pharmacists in veterinary care.

	Animal Hospital Staff		Pharmacy/Drug Store Staff	
	N (%)	Adjusted Residuals	N (%)	Adjusted Residuals
Have you ever thought about pharmacists being involved in companion animal veterinary medicine?				
Yes	124 (57.1)	2.68 **	147 (45.4)	–2.68 **
No	84 (38.7)	−3.00 **	168 (51.9)	3.00 **
Neither option can be selected	9 (4.2)	0.87	9 (2.7)	–0.87
Total	217 (100)		324 (100)	

Note: Responses were compared between the groups using the chi-square test, with the *p*-value for significance set to 0.05. Statistical significance was noted (** *p* < 0.01).

**Table 6 pharmacy-12-00179-t006:** Consideration of future collaboration between veterinary and pharmacy staff.

	Animal Hospital Staff		Pharmacy/Drug Store Staff	
	N (%)	Adjusted Residuals	N (%)	Adjusted Residuals
Would you like to work with a pharmacist/veterinarian or veterinary nurse for companion animals in the future?				
I strongly agree	23 (10.6)	−1.71	51 (15.7)	1.71
If I dare to answer, I think so	58 (26.7)	−2.07 *	114 (35.2)	2.07 *
Neither option can be selected	66 (30.4)	−0.56	106 (32.7)	0.56
If I dare to answer, I do not think so	55 (25.4)	3.37 **	45 (13.9)	–3.37 **
I never think so	15 (6.9)	2.51 *	8 (2.5)	–2.51 *
Total	217 (100)		324 (100)	

Note: Responses were compared between the groups using the chi-square test, with the *p*-value for significance set to 0.05. Statistical significance was noted (* *p* < 0.05; ** *p* < 0.01).

## Data Availability

Data are contained within the article and Appendix A.

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
