# Peer review of "Current Situation for Pharmacists in Japanese Veterinary Medicine: Exploring the Pharmaceutical Needs and Challenges of Veterinary Staff to Facilitate Collaborative Veterinary Care"

_pharmacy, 2024, doi:10.3390/pharmacy12060179_

Round 1
Reviewer 1 Report
Comments and Suggestions for Authors
Konno et al. have written an insightful manuscript on the current role of pharmacists in Japanese veterinary medicine. Overall, the manuscript is well-structured, and the data are presented clearly. However, the main concern is with the figure quality. Presenting figures in black and white hinders comprehension due to the density of information; using color would enhance clarity. Additionally, please consider the following points:
-
It would be better to move the sentence, 'The survey was conducted between May and August 31, 2023,' to the 2.1 Survey section.
-
Including a map of the survey area would greatly aid understanding. Please add a map showing the survey area.
-
The discussion section could be expanded. It would be beneficial to compare the Japanese context with that of other countries, as the current discussion primarily focuses on Japan. For instance, you might include comparisons with other neighboring Asian countries.
Author Response
The authors would like to thank the reviewers for their constructive critique to improve the manuscript. We have made every effort to address the issues raised and to respond to all comments. Please, find next a detailed, point-by-point response to the reviewers’ comments. We hope that our revisions will meet the reviewers’ expectations.
Reviewer1
Comments 1: Konno et al. have written an insightful manuscript on the current role of pharmacists in Japanese veterinary medicine. Overall, the manuscript is well-structured, and the data are presented clearly. However, the main concern is with the figure quality. Presenting figures in black and white hinders comprehension due to the density of information; using color would enhance clarity.
Response 1: The authors sincerely thank you for the valuable suggestions and constructive feedback. In response to the concern regarding figure quality, we have reproduced all figures in full color to enhance clarity and improve comprehension. We believe this adjustment addresses the issue effectively. The updated figures are included in the revised manuscript for your review.
Comments 2: It would be better to move the sentence, 'The survey was conducted between May and August 31, 2023,' to the 2.1 Survey section.
Response 2: We have moved the sentence “The survey was conducted between May 1 and August 31, 2023” to P2 L79.
Comments 3: Including a map of the survey area would greatly aid understanding. Please add a map showing the survey area.
Response 3: Thank you for you helpful suggestion. In response, we have added a map of the surveyed area in ​​Japan, which is included as supplementary material (supplementary material_S3).
Comments 4: The discussion section could be expanded. It would be beneficial to compare the Japanese context with that of other countries, as the current discussion primarily focuses on Japan. For instance, you might include comparisons with other neighboring Asian countries.
Response 4: We agree with these remarks. Hence, we have added text to the Discussion regarding the state of veterinary pharmacy in other countries (P18 L349–355).
Reviewer 2 Report
Comments and Suggestions for Authors
Pharmacy: Review of paper by Konno et al. 2024
Current Situation for Pharmacists in Japanese Veterinary Medi- 2
cine: Exploring the Pharmaceutical Needs and Challenges of 3 Veterinary Staff to Facilitate Collaborative Veterinary Care
The manuscript by Konna is well-written and describes the interesting topic of the collaborative opportunities between pharmacists and vets in Japan. The study collates information obtained from surveys to provide initial insights into the status of veterinary pharmacy in Japan. However, there is minimal comparison with practices in other parts of the world, and this should be added.
It would be valuable if the authors could present the information in Table 2 about the region of Japan on a map that includes the main cities in Japan. This will help international readers to orient themselves with the geographical data presented.
Can the authors to describe the validation method they used before disseminating the surveys?
In Fig. 3, it is difficult to distinguish between the columns of data from “veterinarians” and “general employee”. Could the authors re-do this graph and use colours to distinguish the different groups instead of the current shading patterns?
The use of colour would also enhance the presentation of the other figures in the ms.
The suggestion by the authors that there is opportunity collaboration is important and I worthy suggestion. The manuscript would benefit from some specifics about how / what pharmacists can contribute. In the paragraph that ends on Line 363, this idea is raised, however falls short of giving any details.
Demographic information was collected for the sites surveyed, including the age of the respondents? It was not apparent that any analysis was performed using this data. Could this be performed to understand if there are differences in response with age and/or years of practice. If not, then this data does not need to be reported in the main ms.
The use of English in the responses given in Fig.7 are difficult to interpret.
There is other literature published in this topic area and I suggest that the authors consider incorporating these works into the current manuscript.
Davidson, G. Veterinary Compounding: Regulation, Challenges, and Resources. Pharmaceutics 2017, 9, 5. https://doi.org/10.3390/pharmaceutics9010005 Young NW, Royal KD, Park M, Davidson GS. Pharmacists’ Knowledge of Veterinary Pharmacotherapy and the Impact of an Educational Intervention. Journal of Pharmacy Technology. 2018;34(6):244-251. doi:10.1177/8755122518794023
M. L. Ceresia, C. E. Fasser, J. E. Rush, R. T. Scheife, C. J. Orcutt, D. I. Michalski, et al. The role and education of the veterinary pharmacist. American Journal of Pharmaceutical Education 2009 Vol. 73 Pages 1-10
E.g. Besley N, Browne P, Park M, Pesheva P, Wong K, Medlicott NJ and McDowell A (2023). Pharmacists in zoos? A qualitative study investigating the potential for pharmacist involvement in wildlife healthcare in Aotearoa New Zealand. Journal of American Pharmacists Association and the references cited within.
Specific comments and typographical errors are listed below:
Line 42. We are now in the year 2024 and so the tense of this statement referring to 2023 should be changed from ‘predicts’.
Fig. 1. Caption. The way this is written is a little cumbersome and repetitive. I would suggest revising this text to assist comprehension by the reader.
Line 226. Please revise use of the phrase “…drug picking…”. Perhaps “drug selection” instead?
Line 347. The phrase “owing to the lack of advancement in the pharmaceutical division of labor” requires further explanation. It is unclear what specifically the authors mean by this statement. Please be specific.
Line 381. Why / how does an uneven regional distribution of pharmacists limit the implementation of veterinary pharmacy?
Author Response
The authors would like to thank the reviewers for their constructive critique to improve the manuscript. We have made every effort to address the issues raised and to respond to all comments. Please, find next a detailed, point-by-point response to the reviewers’ comments. We hope that our revisions will meet the reviewers’ expectations.
Reviewer2
Comments 1: The manuscript by Konna is well-written and describes the interesting topic of the collaborative opportunities between pharmacists and vets in Japan. The study collates information obtained from surveys to provide initial insights into the status of veterinary pharmacy in Japan. However, there is minimal comparison with practices in other parts of the world, and this should be added.
Response 1: Thank you very much for reviewing this paper and providing feedback. We agree with these remarks. Hence, we have added a comparison with other countries to the Discussion (P18 L349–355).
Comments 2: It would be valuable if the authors could present the information in Table 2 about the region of Japan on a map that includes the main cities in Japan. This will help international readers to orient themselves with the geographical data presented.
Response 2: Thank you for your helpful suggestion. In response, we have added a map of the surveyed area in ​​Japan, which is included as supplementary material (supplementary material_S3).
Comments 3: Can the authors to describe the validation method they used before disseminating the surveys?
Response 3: Thank you for your insightful comment. The questionnaire used in this study was developed based on insights from previous studies conducted in Japan and internationally. It was specifically designed to provide foundational knowledge for Japanese pharmacists aiming to work in the veterinary medical field. While sensitivity and specificity tests were not conducted on the developed questionnaire. The authors collaboratively determined the survey subject selection method, response rate estimation, sample size, and refinement of the questionnaire. Furthermore, the content of the questions was reviewed and approved by the Ethics Review Committee of Tohoku Medical and Pharmaceutical University (Ethics Review Number: 2023-0-001). The manuscript has been updated to include this information (P2, L78–79).
Comments 4: In Fig. 3, it is difficult to distinguish between the columns of data from “veterinarians” and “general employee”. Could the authors re-do this graph and use colours to distinguish the different groups instead of the current shading patterns?
Response 4: Thank you for pointing this out. Figure 3 is now color-coded by group, and the numbers of participants are also provided to make it easier to identify items with few respondents (P8).
Comments 5: The use of colour would also enhance the presentation of the other figures in the ms.
Response 5: All figures have been reproduced in full color.
Comments 6: The suggestion by the authors that there is opportunity collaboration is important and I worthy suggestion. The manuscript would benefit from some specifics about how / what pharmacists can contribute. In the paragraph that ends on Line 363, this idea is raised, however falls short of giving any details.
Response 6: Thank you for pointing this out. To address the comment, we have included specific examples of how pharmacists can contribute to collaboration with veterinary professionals. For instance the pharmacists can optimize drug therapy by compounding medications that are more palatable for pets and providing them in forms that are easier for owners to administer. This collaboration could also help reduce the economic burden on pet owners and improving convenience in certain cases. An explanation of these concepts has been added (P18-19 L355–363).
Comments 7: Demographic information was collected for the sites surveyed, including the age of the respondents? It was not apparent that any analysis was performed using this data. Could this be performed to understand if there are differences in response with age and/or years of practice. If not, then this data does not need to be reported in the main ms.
Response 7: Thank you for your valuable comments. Following your suggestion, we have created a new table including data on age and length of service and have included it in the supplementary materials (Supplementary_Material_Respondent_Characteristics_S4). Additionally, we have recreated the tables using the remaining data (Tables 1–3: p3 L127–P5 L132) and have revised the manuscript accordingly (Manuscript: P3 L118–L123). If you require any additional analyses, please provide the details for our consideration.
Comments 8: The use of English in the responses given in Fig.7 are difficult to interpret.
Response 8: Thank you for pointing this out. The English in the answer options in Figure 7 has been corrected (p15), Along with this correction, the supplementary file has also been corrected (Supplementary_Material_Questionnaire_S1_pharmacy_R1, P5–6).
Comments 9: There is other literature published in this topic area and I suggest that the authors consider incorporating these works into the current manuscript.
Davidson, G. Veterinary Compounding: Regulation, Challenges, and Resources. Pharmaceutics 2017, 9, 5. https://doi.org/10.3390/pharmaceutics9010005 Young NW, Royal KD, Park M, Davidson GS. Pharmacists’ Knowledge of Veterinary Pharmacotherapy and the Impact of an Educational Intervention. Journal of Pharmacy Technology. 2018;34(6):244-251. doi:10.1177/8755122518794023
- L. Ceresia, C. E. Fasser, J. E. Rush, R. T. Scheife, C. J. Orcutt, D. I. Michalski, et al. The role and education of the veterinary pharmacist. American Journal of Pharmaceutical Education 2009 Vol. 73 Pages 1-10
E.g. Besley N, Browne P, Park M, Pesheva P, Wong K, Medlicott NJ and McDowell A (2023). Pharmacists in zoos? A qualitative study investigating the potential for pharmacist involvement in wildlife healthcare in Aotearoa New Zealand. Journal of American Pharmacists Association and the references cited within.
Response 9: We thank you for this suggestion. The references provided by the reviewers were beneficial for improving this manuscript. We have included them to better illustrate the situation in other countries and strengthen the Discussion section (P18–21 L331–333, L349–363).
Comments 10: Specific comments and typographical errors are listed below:
Line 42. We are now in the year 2024 and so the tense of this statement referring to 2023 should be changed from ‘predicts’.
Response 10: Thank you for pointing this out. We have revised accordingly (L42).
Comments 11: Fig. 1. Caption. The way this is written is a little cumbersome and repetitive. I would suggest revising this text to assist comprehension by the reader.
Response 11: Thank you for your careful review. We have revised as suggested (L166–170).
Comments 12: Line 226. Please revise use of the phrase “…drug picking…”. Perhaps “drug selection” instead?
Response 12: In order to maintain a formal tone and improve the clarity of our intended meaning, we have changed “drug picking” to “preparing the drugs with a prescription” in L222.
Comments 13: Line 347. The phrase “owing to the lack of advancement in the pharmaceutical division of labor” requires further explanation. It is unclear what specifically the authors mean by this statement. Please be specific.
Response 13: Thank you for pointing this out. Our intent was not properly expressed. Here, “the pharmaceutical division of labor” means “the separation of prescription and dispensing activities.” The manuscript has been revised accordingly (P18 L345–346).
Comments 14: Line 381. Why / how does an uneven regional distribution of pharmacists limit the implementation of veterinary pharmacy?
Response 14: Thank you for your comment and for bringing this point to our attention. In Japan, pharmacists are expected to play an active role not only in dispensing medicines in the narrow sense (e.g., preparing the drugs with a prescription) but also in optimizing drug therapy for humans. This responsibility often leads to a concentration of pharmacists in urban areas, where demand is higher. In regions with fewer pharmacists, there is a tendency to prioritize human healthcare over veterinary needs, which limits the implementation of veterinary pharmacy. While addressing the uneven regional distribution of pharmacists remains a critical issue within Japan’s human medical system, veterinary pharmacy remains an area that requires focused efforts to support its development. The manuscript has been revised to ensure this point is accurately conveyed to readers (P19 L386–388).
Round 2
Reviewer 2 Report
Comments and Suggestions for Authors
The author's have addressed my comments to my satisfaction.